# Using Steady-State Kinetics to Quantitate Substrate Selectivity and Specificity: A Case Study with Two Human Transaminases

**DOI:** 10.3390/molecules27041398

**Published:** 2022-02-18

**Authors:** Alessio Peracchi, Eugenia Polverini

**Affiliations:** 1Department of Chemistry, Life Sciences and Environmental Sustainability, University of Parma, 43124 Parma, Italy; 2Department of Mathematical, Physical and Computer Sciences, University of Parma, 43124 Parma, Italy; eugenia.polverini@unipr.it

**Keywords:** substrate specificity, molecular discrimination, limits of specificity, enzyme assays, transaminases, enzyme evolution

## Abstract

We examined the ability of two human cytosolic transaminases, aspartate aminotransferase (GOT1) and alanine aminotransferase (GPT), to transform their preferred substrates whilst discriminating against similar metabolites. This offers an opportunity to survey our current understanding of enzyme selectivity and specificity in a biological context. Substrate selectivity can be quantitated based on the ratio of the *k_cat_*/*K_M_* values for two alternative substrates (the ‘discrimination index’). After assessing the advantages, implications and limits of this index, we analyzed the reactions of GOT1 and GPT with alternative substrates that are metabolically available and show limited structural differences with respect to the preferred substrates. The transaminases’ observed selectivities were remarkably high. In particular, GOT1 reacted ~10^6^-fold less efficiently when the side-chain carboxylate of the ’physiological’ substrates (aspartate and glutamate) was replaced by an amido group (asparagine and glutamine). This represents a current empirical limit of discrimination associated with this chemical difference. The structural basis of GOT1 selectivity was addressed through substrate docking simulations, which highlighted the importance of electrostatic interactions and proper substrate positioning in the active site. We briefly discuss the biological implications of these results and the possibility of using *k_cat_*/*K_M_* values to derive a global measure of enzyme specificity.

## 1. Introduction

In introductory textbooks of biochemistry, substrate specificity is often described as a defining feature of enzymes. Yet the significance of the term is often left implicit and a quantitative description of specificity is not offered. Intuitively, specificity represents the capacity of an enzyme to bind (and transform) a particular substrate while avoiding other, chemically similar, molecules [1]. In a cellular setting, specificity entails reacting with a single substrate in preference to a multitude of other available metabolites. To the host organism, a highly specific enzyme may offer several advantages, particularly if it is involved in some central metabolic pathway: specialized enzymes permit a finer regulation of metabolism and prevent interferences between different pathways, while also limiting the formation of unwanted metabolic ‘junk’ [2].

It is a fact that metabolic enzymes are often highly selective towards a single substrate. Nonetheless, it is also well appreciated that a perfect discrimination against other, similar molecules is essentially unattainable [3,4]. Furthermore, different orthologous (isofunctional) enzymes will show different degrees of reactivity towards alternative substrates [5]. This all means that specificity must be thought of as a quantitative, rather than a qualitative trait. 

Accordingly, finding the proper and meaningful way to measure and express specificity seems interesting in many regards. For example, it may help properly assess how well an enzyme discriminates between two substrates that show a specific chemical difference, as compared to other enzymes facing the same problem [4]. If structural data are available, it may allow the active site features to be connected to the degree of discrimination shown by that specific enzyme [6]. Finally, there are hints that there may be a tradeoff between specificity and overall catalytic efficiency [7], but testing this possibility requires some rigorous way to express specificity.

All of this implies that there must be a quantitative measure of specificity, and elicits the question as to how it can be measured. The answer is, necessarily, through the establishment of an enzyme’s steady-state catalytic parameters. While for quite a long time different authors disagreed as to which catalytic parameter (*k_cat_*, *K_M_*, or *k_cat_*/*K_M_*) was most useful for comparisons aimed at defining specificity [8], today it is agreed that discrimination between two competing substrates is best expressed by comparing the respective *k_cat_*/*K_M_* values [9]. Using two distinct approaches, Fersht [3] and Cornish-Bowden [10] derived an equation describing the kinetics of an enzyme that acts simultaneously on two competing substrates (A and B) whose respective *K_M_* values equal the *K_i_* values for the other reaction. When the two reactions occur simultaneously at rates v_A_ and v_B_, respectively, the following relationship holds:(1)$$\frac{v_{A}}{v_{B}}=\frac{\left(k_{\mathrm{cat}}^{A}/K_{M}^{A}\right)}{\left(k_{\mathrm{cat}}^{B}/K_{M}^{B}\right)}\times \frac{\left[A\right]}{\left[B\right]}$$

In the lab practice, kinetic parameters can be established from measurements carried out with each substrate separately, and Equation (1) can thus be used to compare enzyme rates at any concentrations of the two competing substrates.

Equation (1) implies that, in the presence of equimolar concentrations of the two competing substrates, the ratio of rates for the two reactions will be equal to the ratio of *k_cat_*/*K_M_* values, no matter whether the enzyme is saturated or not. It makes sense therefore to use this ratio as a measure of the discrimination between the two substrates. This ratio is called the ‘discrimination factor’ or ‘discrimination index’ and is hereby indicated by the letter D: (2)$$D\;=\frac{\left(k_{\mathrm{cat}}^{A}/K_{M}^{A}\right)}{\left(k_{\mathrm{cat}}^{B}/K_{M}^{B}\right)}$$

The property described quantitatively by the index D is usually called selectivity (or accuracy) [7] and it will be systematically employed in this work. Even though Equation (2) is valid for any given couple of substrates, irrespective of the order in which they are compared, within this article, it is assumed that substrate A is the preferred one, while substrate B is the alternative (less reactive) substrate. By this convention, the values of D can range between 1 and +∞.

Note that the term specificity, as mentioned earlier, has a broader meaning than selectivity, referring not just to the discrimination by the enzyme of one substrate *versus* another, but rather to the systematic preference towards one (or a few) substrates with respect to all other compounds. However, while selectivity may be quantitated quite satisfactorily through the D index, there is no accepted synthetic measure of specificity.

The present paper analyzes some aspects of the selectivity and specificity of two human transaminases—the cytosolic aspartate aminotransferase (GOT1) and the cytosolic alanine aminotransferase (GPT). We will discuss how enzyme kinetic data can be properly employed to calculate the discrimination indexes for these enzymes, comparing them with those shown by other classes of enzymes and finally assessing the boundaries of selectivity in the presence of some precise chemical differences between the preferred and alternative substrates. 

## 2. Results

### 2.1. Defining the Appropriate Measurement and Use of Kinetic Parameters to Analyze the Selectivity of Transaminases

Aminotransferases (transaminases) are enzymes that catalyze the transfer of the amino group from a donor (most commonly, an α-amino acid) to an acceptor that bears a carbonyl function (typically, an α-keto acid). For example, the two transaminases analyzed in this study, GOT1 and GPT, carry out the ‘canonical’ reactions outlined in Figure 1. 

These transaminases adopt a ping-pong kynetic mechanism. The reactions in Figure 1 (and those of most other aminotransferases as well) are fully reversible, and may take place in either direction under in vivo conditions. Accordingly, an aminotransferase must be able to recognize and transform two distinct amino acids (e.g., in the case of GPT, l-alanine and l-glutamate), which are necessarily different in terms of shape and side-chain chemical properties. The molecular mechanisms that underlie dual substrate recognition have been described for various classes of transaminases [11,12]. In any case, active sites that are capable of binding two distinct substrates may have an inherent difficulty in discriminating against many other amino acids that are present in the cell. 

As stated in the introduction, substrate selectivity is generally evaluated by taking in consideration the parameter *k_cat_*/*K_M_*, from which ultimately the discrimination factor (D) can be calculated. There are however limits, caveats and implications in this approach that deserve to be addressed before performing the kinetic measurements and analyzing their outcome.

Can we properly calculate selectivity for the given enzymes? Strictly speaking, *k_cat_*/*K_M_* is a parameter that can be determined only for enzymes that show Michaelis–Menten behavior (i.e., hyperbolic kinetics). It has been argued that the treatment of non-Michaelis–Menten enzymes is also possible, but it is certainly less straightforward [8]. In any case both GOT1 and GPT follow Michaelis–Menten kinetics (i.e., they do not show appreciable cooperativity) so there is no a priori objection to extracting a proper value of *k_cat_*/*K_M_* from their kinetics.

These enzymes require a co-substrate: at which concentration should it be kept? Indeed, transaminases catalyze reactions between two susbtrates—a keto acid and an amino acid (Figure 1). In these instances, in most enzymologycal studies, *k_cat_*/*K_M_* for one substrate is obtained in the presence of a saturating concentration of the other. However this condition is not always achievable and, more importantly, it is not physiologically realistic, since in vivo enzymes are seldom if ever saturated by their substrates. Cornish-Bowden showed convincingly that, as long as one is concerned with specificity in a physiological setting, *k_cat_*/*K_M_* for one substrate should be measured at some concentration of the other substrate that approaches the in vivo concentration (for the sake of rigorous terminology, the value of the *k_cat_*/*K_M_* parameter obtained under such conditions will be deemed ‘apparent’) [13]. Accordingly, in this study, the initial concentrations of the compounds used as co-substrates were close to the reported concentrations of the same compounds in the cytosol of mammalian cells. For example, pyruvate was used at a fixed concentration of 250 μM [14] and l-glutamate at 5 mM [15]. The only exception was oxaloacetate, which for practical reason was used in assays at a concentration (200 μM) that is presumably one order of magnitude greater than the concentration in vivo.

Which alternative substrates may be more meaningful to test? In principle, selectivity can be tested by comparing *k_cat_*/*K_M_* of the preferred substrate with that of any alternative substrate, either natural or synthetic. The latter kind of substrates may be interesting in an applicative perspective. However, if we are trying to learn about the ‘physiological’ properties of an enzyme, and maybe about the selective pressures that have shaped its specificity, it seems worth focusing on substrates that are normally present in the cell. Here, we considered amino acids (such as l-asparagine and l-glutamine) and keto acids (α-ketobutyrate) that are intermediates of primary metabolism. 

What about substrates that bind weakly (failing to reach saturation) or that show substrate inhibition? Quite conveniently, if an enzyme shows hyperbolic kinetics, the *k_cat_*/*K_M_* parameter (or, as in our case, the apparent *k_cat_*/*K_M_*) is the easiest to establish. In fact, *k_cat_*/*K_M_* represents the slope of the initial part of the Michaelis–Menten hyperbola (reaction rate vs. substrate concentration). Therefore, even for those enzyme-substrate couples in which substrate saturation cannot be achieved due to poor binding (hence preventing the attainment of precise individual values for *k_cat_* and *K_M_*), one can obtain a reliable estimate of their ratio. Similarly, for substrates that show substrate inhibition (a phenomenon very common in enzymes that follow a ping-pong mechanism, notably transaminases) the slope of the earliest part of the titration curve will still represent the correct *k_cat_*/*K_M_* to an excellent approximation.

The discrimination index (D) does not distinguish between selectivity at the substrate binding stage and selectivity in substrate transformation—or does it? Discrimination can mirror both reduced binding affinity and reduced reactivity of the substrate once bound. Both aspects are encompassed in *k_cat_*/*K_M_*. In particular, to the extent that transition state theory is applicable to the type of reaction under examination, *k_cat_*/*K_M_* is directly related to the energy difference between the free substrate (and free enzyme) and the enzyme-bound transition state. This is formally demonstrated, for a very simple kinetic scenario, in Appendix A. In this simple scenario, the discrimination index will depend on the differential binding of the two substrates, again in the transition state (Appendix A). Hints as to whether discrimination is partly exercised at the initial binding stage (formation of the enzyme-substrate complex) can be obtained when the *K_M_* values for the competing substrates can be reliably measured. 

### 2.2. Assessing the Discrimination of Human GOT1 and GPT against a Selected Subset of Alternative Substrates

The activity of GOT1 and GPT was measured using the preferred substrates (l-glutamate and l-aspartate for GOT1; l-glutamate, l-alanine and pyruvate for GPT), as well as a selected subset of alternative substrates that are present in the cell and show small and well-defined structural differences with respect to the preferred substrates. The assays were conducted under pseudophysiological conditions (pH 7.4, 37 °C) using a series of coupled assays (some continuous and other discontinuous) described in the Methods. 

The results for GOT1 are summarized in Table 1. As it can be seen, the enzyme discriminated both l-asparagine and l-alanine (which do not contain a carboxylate group in the side chain) by about 10^6^ fold with respect to l-aspartate; analogously, l-glutamine was discriminated ~10^6^ fold with respect to l-glutamate. While the discrimination towards l-asparagine and l-alanine showed up both in terms of *k_cat_*/*K_M_* and of *K_M_*, replacement of the γ−carboxylate group of l-glutamate with the amido group of l-glutamine left *K_M_* virtually unchanged, suggesting that the discrimination was occurring almost solely at the catalytic step. The discrimination shown by GOT1 towards l-glutamine and l-alanine was significantly higher than the discrimination towards the same compounds reported for aspartate aminotransferases from other sources [16,17,18]. 

The results obtained with GPT are summarized in Table 2. In general, the discrimination indexes shown by this enzyme were significantly lower than those shown by GOT1. 

In particular, the discrimination associated with replacement of the sidechain carboxylate of glutamate with an amido group (l-glutamate vs. l-glutamine comparison) was of ‘only’ four orders of magnitude (rather than six as seen in GOT1). In all cases, the modified substrates showed an increased *K_M_*, consistent with selectivity occurring at least in part at the initial substrate binding step.

### 2.3. Putting Data in a Context and Assessing the Empirical Limits of Discrimination for Specific Chemical Differences

To put the experimental data in a context and better appreciate the significance of the D values calculated for GOT1 and GPT, such values were compared with the selectivities of other enzymes, retrieved from the literature. The analysis focused on cases where enzymes had to discriminate between substrates showing the same chemical differences explored for the GOT1 and GPT substrates.

In particular, Figure 1 shows the comparative ability of different enzymes to discriminate between a preferred substrate bearing a carboxylate group (e.g., aspartate) and an alternative substrate where the carboxylate is replaced by an amido group (e.g., asparagine). It can be seen that the D values shown by the two human transaminases lie near to (in the case of GPT) or higher than (GOT1) the highest discrimination factors known for other enzymes. 

It is unfortunate that the data for GOT1 and GPT cannot be properly compared with data from Asp or Glu aminocyl-tRNA transferases, as in general aminocyl-tRNA transferases represent a benchmark of substrate selectivity [20]. Yet it is worth noting that the 10^6^-fold discrimination indices shown by GOT1 towards l-asparagine and l-glutamine are above the highest discrimination levels shown by aminoacyl-tRNA synthetases [20].

A second comparison, shown in Figure 2, regards couples in which the reactivity with a preferred substrate bearing a carboxylate group (e.g., aspartate) is contrasted with the reactivity of some analog in which the carboxylate group is replaced by a proton (e.g., alanine). Even though the number of comparisons is limited, once again the discrimination shown by GOT1 is among the highest in the plot.

Finally, Figure 3 compares the discrimination of GPT against l-aspartate with the selectivities of other enzymes acting on a preferred linear substrate against a competitor shorter by one methylene group. The D value shown by GPT is again among the highest ones, comparable to the discrimination of *E. coli* leucyl-tRNA synthetase against l-valine.

### 2.4. Docking Simulations of Substrates into the GOT1 Active Site

One of the most striking observations from the previous two sections is the very strong discrimination shown by GOT1 towards substrates containing an amido group. To gain insights into the chemical and structural basis of this discrimination, we performed docking simulations on the substrate-binding site of this enzyme. 

The features of the GOT1 active site allow it to favorably accommodate the cofactor pyridoxal 5′-phosphate (PLP) and its substrate(s), forming with them a plethora of H-bonds and ionic interactions. Inspection of the homologous GOT1 from pig (PDB ID: 5VJZ, in which the PLP cofactor forms a Schiff base with the substrate analog α-methylaspartate [21]) helped us to pinpoint the key residues of the binding site. These include several charged residues, in particular arginines, and a tryptophan (Trp141) that stacks onto the pyridoxal ring (Figure 4) and is also able to donate an H-bond to the substrate side chain. Such binding site residues belong to both chains of the dimer. The dimeric form of GOT1 is, in fact, crucial for substrate binding, owing in particular to the presence of an arginine residue from the other monomer (Arg293, human GOT1 sequence numbering) that forms a salt bridge with the sidechain carboxylate group of the substrate (Figure 4b). 

When we computationally docked different substrates in the GOT1 active site, we found that this key salt bridge was preserved in all the docked conformations of the acidic substrates (l-aspartate and l-glutamate), which recover an orientation into the active site that strictly resembles the one of methylaspartate-PLP in the pig enzyme [21] (Figure 4 and Appendix A). With the cluster analysis, however, another conformation of the ligand is found, even if at higher energy and very low incidence. Such a conformation presents a 180° rotation around α-carbon that leads the sidechain carboxylate to interact with Arg387 (which normally binds the α-carboxylate group) instead of Arg293 (Appendix A). This “rotated” binding mode becomes relevant when asparagine- or glutamine-PLP are docked in the binding site. In fact, the substitution of the sidechain carboxylate with an amido group eliminates the strong electrostatic interaction with Arg293, as well as the H-bond interaction with Trp141. As a consequence, not only do l-asparagine and l-glutamine show a much worse binding energy (compared to their carboxylate counterparts) but also, for these substrates, both the ‘canonical’ and ‘rotated’ conformation become equally favored (Appendix A).

The position of the Lys259, involved in the reaction mechanism, is also relevant. In the canonical ligand conformation, the ε-amino group of this residue points towards the Cα of the substrate, in a position compatible with a catalytic proton transfer. In the rotated binding mode, this becomes impossible.

### 2.5. GPT Model and Comparison with GOT1 Binding Site

The dimeric structure of human cytosolic alanine aminotransferase built by the Swiss-Model (see the Materials and Methods section) was selected for the analysis of the binding site and for its comparison with the GOT1 one. Unfortunately, a modeled structure is unsuitable for docking simulations, due to the low resolution of its atomic positions. The orientation of the binding site side chains is, in fact, crucial for the correct positioning and orientation of the ligand. However, valuable information can be obtained about the active site residues directly involved in substrate binding, in particular, by comparing the modelled structure of GPT with that of GOT1 docked with Glu-PLP, which is also a GPT substrate.

The site residues that in GOT1 form H-bonds or ionic bonds with the substrate (Figure 4) are also very conserved in GPT (Appendix A), supporting the reliability of the model. Additionally, orientation of these residues is in agreement with their expected functional role. In particular, the Lys residue involved in proton transfer, the two Arg that form ionic bonds with the phosphate group and with the α-carboxylate, and the Asp that interacts with the protonated nitrogen of the pyridoxal ring are retained. In addition, the stacking of an aromatic residue with the PLP ring is preserved: Trp141 of GOT1 is substituted in GPT by Tyr189 (in the same orientation), which, with its OH, could donate an H-bond to the side chain of a polar substrate (Appendix A). However, a remarkable difference involves the residue mostly responsible for binding the substrate side chain. In fact, the crucial Arg293 present in GOT1 is functionally replaced in GPT by a tyrosine residue, Tyr35, which belongs to the same monomer as the catalytic Lys and points its OH group towards the substrate. This substitution of a positively charged residue with a polar one (which can form at most a H-bond, rather than a salt bridge, with the substrate sidechain carboxylate) can heavily affect the strength of the binding. Another distinctive feature in the active site of GPT is the presence of a leucine (Leu348, belonging to the other monomer), which is also oriented towards the substrate side chain. Its occurrence in GPT may serve to help the binding of l-alanine, by creating a favorable hydrophobic environment for its –CH_3_ group (Appendix A).

## 3. Discussion

### 3.1. Selectivity: A Case Study with Two Efficient Transaminases

In all evidence, the ability of an enzyme to discriminate between substrates depends on a mix of physico-chemical and biological (evolutionary) factors [4,20,22]. Chemistry somehow sets the boundaries of selectivity—boundaries that are in general difficult to estimate beforehand, but result from the finite differences in binding energies associated with the structural differences between substrates [4].

However, within the limits imposed by chemistry, the actual degree of selectivity of an enzyme is largely shaped by natural selection [4,23]. The classic example is aminoacyl-tRNA synthetases: for these enzymes, their crucial role in protein synthesis requires them to achieve a high reactivity towards only one amino acid and to strongly discriminate against the others [20,24]. Oftentimes, the selectivity exerted by these aminoacyl-tRNA synthetases is higher than that of any other enzyme faced with the same type of discrimination problem [20], so that it can be considered a ‘benchmark’ or an empirical limit of selectivity.

Aspartate aminotransferase (GOT1) and alanine aminotransferase (GPT) are among the most expressed transaminases in many tissues (including liver), where they play key roles in amino acid metabolism. Here they have been used as a case study for the analysis of selectivity and specificity. Although GOT1 and GPT have always been considered very specific, the data herein confirm this opinion quantitatively, by assessing the selectivity of these enzymes against physiologically available alternative substrates. Notably GOT1 and GPT are not only very selective in absolute terms, but also in relative terms, as compared to other enzymes, including aminoacyl-tRNA synthetases (Figure 1, Figure 2 and Figure 3). The selectivity is so high, in fact, that the preference of GOT1 towards l-aspartate and l-glutamate as compared to their amido counterparts (l-asparagine and l-glutamine) sets a current empirical limit for the enzymic discrimination towards this type of chemical difference (Figure 1).

In addition to their outstanding selectivity, GOT1 and GPT also show an excellent catalytic efficiency, with apparent *k_cat_* values, for example, above the average for central metabolic enzymes [25]. Such an efficiency is even more remarkable considering the complex ping-pong mechanism of transaminases, in which two substrates must alternatively bind to the active site during the catalytic cycle, and form various pyridoxal-phosphate-bound intermediates. This combination of high efficiency and very high selectivity may represent an exception to the frequent trade-off, noted by Tawfik, between selectivity and catalytic performance [7].

### 3.2. Chemical and Biological Factors behind the Selectivity of GOT1 and GPT

In analogy with the case of aminoacyl-tRNA synthetases, for metabolic enzymes, too, the degree to which reactivity with a given alternative substrate is suppressed must depend on the biological burden associated with the corresponding side reaction; however, other factors may be at play. For example, exclusion from the active site compounds that are structurally very different from the preferred substrate should be easier than discriminating against alternative substrates that resemble more the preferred one. The somewhat lower selectivity shown by GPT as opposed by GOT1 is in apparent agreement with this hypothesis.

Indeed, both the preferred substrates of GOT1 (l-glutamate and l-aspartate, if we consider the amino group donor; or a-KG and oxaloacetate, if we consider the amino group acceptors) are characterized by the presence of a carboxylate group in the side chain. Classic aspartate aminotransferases (including GOT1) bind the sidechain carboxylate through an Arg residue (Arg293 in the case of the human GOT1 enzyme), quite rigidly positioned in the substrate binding site [18] (Figure 4). The electrostatic interaction with this residue helps explain not only binding of the preferred substrates’ carboxylate, but also discrimination against alternative substrates bearing the isosteric (but uncharged) amido group. As suggested by our docking simulations (Appendix A), reactivity of GOT1 towards l-asparagine and l-glutamine is thwarted by at least three factors: (i) The substitution of the sidechain carboxylate with an amido group breaks the electrostatic interaction of the substrate with Arg293, weakening binding; (ii) The ensuing repositioning of the side chain of the substrate favors a small but perceivable movement of the α-carbon, which may hinder the catalytic proton transfer events that have to occur at this carbon during the transamination reaction; (iii) Finally, substrates that lack the side-chain carboxylate can much more easily adopt a ‘rotated’, clearly unproductive, binding mode in the active site. While the occurrence of such an alternative binding mode is not expected, to a first approximation, to affect the *k_cat_*/*K_M_* parameter, it may nevertheless reduce the reactivities of GOT1 towards l-glutamine and l-asparagine at high substrate concentrations.

The discrimination task may be a bit more complex in the case of GPT, where one preferred substrate (alanine or pyruvate) does not include any negatively charged group in the side chain. Indeed, the structural alignment of the GPT model with the GOT1 structure shows that the GOT1′s Arg293 is replaced by a Leu residue in GPT (Appendix A). Accordingly, the sidechain carboxylate of the l-glutamate substrate seems unable to interact with GPT through a salt bridge. On the other hand, inspection of the structural model of the enzyme active site suggests that the carboxylate might be forming a hydrogen bond with Tyr35 (Appendix A; according to both the Uniprot database [26] and the B6 Database [27], this Tyr is strictly conserved in eukaryotic alanine aminotransferases). These differences may render the discrimination against substrates with an uncharged side chain, e.g., l-glutamine, less straightforward for GPT (Figure 2). 

Even with this distinction between GOT1 and GPT, selectivity against l-glutamine and l-asparagine by both enzymes is very high—a behavior that seems to make biological sense. In fact, transamination of these amido-containing amino acids is substantially irreversible [28,29] and is usually coupled to processes that need to be thermodynamically driven [30]. Therefore, transamination of l-glutamine and l-asparagine through a side reaction by two abundant transaminases would represent an objective waste of resources that evolution might tend to restrict. 

### 3.3. How Can We Step from Selectivity to Specificity?

All of the quantitative analyses described above deal with selectivity, i.e., with the capacity of an enzyme to distinguish between just two substrates. As noted in the introduction, specificity is usually intended to have a broader meaning, referring to the ability of an enzyme to transform one (or a few) particular substrate(s) in preference to all other compounds. However contrary to selectivity (that can be effectively summarized by the D index), there is no accepted synthetic parameter to express specificity. Put otherwise, while the data in Table 1 and Table 2 undoubtedly attest to the high specificity of GOT1 and GPT, such specificity remains described in qualitative terms. 

One possible approach to convert selectivity data into a synthetic measure of specificity could be analogous to that proposed by Nath and Atkins [31]. They elaborated a ‘promiscuity index’ (promiscuity can be seen schematically as the opposite of specificity, even though the proper use of the term is quite debated [4,32,33]), which related the *k_cat_*/*K_M_* parameter of an enzyme towards a given substrate with the sum of *k_cat_*/*K_M_* values towards a number of alternative substrates. Quite similarly, one could compare the reactivity of the preferred substrate with a set of N alternative substrates to calculate a ‘cumulative’ discrimination index (D_N_) (Equation (3)):(3)$$D_{N}=\frac{\left(k_{\mathrm{cat}}^{A}/K_{M}^{A}\right)}{\sum_{i=1}^{N}\left(k_{\mathrm{cat}}^{i}/K_{M}^{i}\right)}$$
where A denotes the preferred substrate and i denotes the i^nth^ alternative substrate. A “relative” discrimination parameter (R_N_), falling within a finite range, can be calculated as follows:(4)$$R_{N}=\frac{\left(k_{\mathrm{cat}}^{A}/K_{M}^{A}\right)}{\left(k_{\mathrm{cat}}^{A}/K_{M}^{A}\right)+\sum_{i=1}^{N}\left(k_{\mathrm{cat}}^{i}/K_{M}^{i}\right)}=\frac{D_{N}}{D_{N}+1}$$
R_N_ is expected to range between 1/(N + 1) (an enzyme that accepts with equal efficiency all the substrates under examination) and 1 (a hypothetical, perfectly specific enzyme). Eventually, the relative discrimination can be placed on a per mille scale ($R_{N}^{‰}$).
(5)$$R_{N}^{‰}=\left(R_{N}-\frac{1}{N+1}\right)\times \frac{N+1}{N}\times 1000$$
$R_{N}^{‰}$ would range between 0 (perfectly non-specific enzyme) and 1000 (a perfectly specific enzyme). For example, in the case of GOT1, comparing the reactivity with l-aspartate against those with l-asparagine and l-alanine, one can calculate a D_2_ value of 6.18 × 10^5^ and a $R_{2}^{‰}$ value of 999.999. For GPT, comparison of the preferred substrate l-glutamate with l-glutamine and l-aspartate yields D_2_ = 1836 and $R_{2}^{‰}$ = 999.18.

However, calculating similar parameters may be little more than an academic exercise. To be both operationally useful and informative about the biological function of an enzyme, any synthetic parameter describing specificity would have to deal with a number of issues, namely:(1)The parameter will necessarily change depending on how many alternative substrates are considered. Assuming that such a parameter may legitimately serve to compare the specificities of different enzymes, the number of substrates should be agreed upon a priori.(2)The choice of the substrates to be compared also seems crucial. In a biological perspective, comparisons should only involve substrates that are physiologically available.(3)It makes sense to consider primarily alternative substrates that are structurally/chemically most similar to the preferred substrate. To this end, the potential alternative substrates should be ranked in order of ‘similarity’ to the preferred one, an operation that necessarily includes some degree of arbitrariness [31].(4)The analysis should take into account the fact that some enzymes utilize more than one preferred substrate (as in the case of transaminases).(5)Finally, a thorough evaluation of specificity (from a biological and evolutionary viewpoint) should also take into account the discrimination against metabolites that are not substrates *stricto sensu* but may act as competitive inhibitors and/or irreversible inactivators of the enzyme under examination. In these cases, evidently, *k_cat_*/*K_M_* cannot be used as a parameter for comparison.

### 3.4. Final Remarks

As a case study in enzyme selectivity and specificity, we have examined the capacity of two common human transaminases (the cytosolic enzymes GOT1 and GPT) to discriminate between their ’physiological’ substrates and other, structurally similar metabolites. 

After establishing (based on a number of arguments and precedents) that an apparent *k_cat_*/*K_M_* is the most useful kinetic parameter to compare the reactivity of these transaminases towards different potential substrates, we tested the ability of GOT1 and GPT to react with a number of metabolites that show small and well-defined structural differences with respect to the preferred substrates. Quantitatively supporting previous conceptions about these enzymes, the observed selectivities of GOT1 and GPT were remarkably high.

We then compared the ‘D’ values shown by GOT1 and GPT with those reported for other enzymes that had been challenged with alternative substrates carrying the same type of chemical differences. This analysis served to highlight, in particular, the extremely high discrimination of GOT1 towards substrates containing an amido group in their side chain in place of a carboxylate. The discrimination indexes shown by GOT1 in these cases (i.e., in the glutamate vs. glutamine and aspartate vs. asparagine comparisons) set the current empirical limit for the ability of an enzyme to distinguish between a carboxylic group and an amide.

Finally, we explored the structural basis of this selectivity, by simulating the docking of different substrates at the GOT1 active site. These simulations indicate that the electrostatic interactions formed by the side-chain carboxylates of aspartate and glutamate are important both for binding and for correct positioning of the substrates in the enzyme active site. Modelling studies on GPT pinpointed some structural features, never described before, of the active site of this enzyme and suggested plausible explanations for its comparatively lower ability to discriminate against glutamine. The lower discrimination shown by GPT appears associated to the presence of two distinctive, uncharged active site residues (Tyr35 and Leu348) that can alternatively substitute GOT1′s Arg293 in the interaction with the substrate side chain.

## 4. Materials and Methods

### 4.1. Enzymes and Chemicals

Bovine liver glutamate dehydrogenase (GDH) was purchased from Roche (Mannheim, Germany). Oxaloacetate was from Boehringer. All other reagents were from Fluka or Sigma–Aldrich (now Merck, Darmstadt, Germany).

The cloning and expression of the human GOT1 and GPT has been described before [34]. The expression clone of human lactate dehydrogenase A (LDHA) was a kind gift from Dr. Lisa Craig (Simon Fraser University, Burnaby, Canada) [35]. The coding sequence of the human cytosolic malate dehydrogenase (MDH1) was cloned in a pET28a vector and the recombinant protein was expressed in *E. coli* BL21(DE) cells; details on the cloning and expression procedure will be provided in an upcoming paper (Caligiore et al., manuscript in preparation). Recombinant ω-amidase from yeast (product of the yNit3 gene) was obtained as described [36]. 

Recombinant hexahistidine-tagged proteins were purified by Ni- or Co-affinity chromatography [34,36]. The enzymes were then transferred to an appropriate storage buffer (typically 50 mM Hepes pH 7.5, 100 mM NaCl, 0.5 mM EDTA, 1 mM DTT, 10% glycerol; for the transaminases, the buffer was also supplemented with 5 μM pyridoxal 5′-phosphate-PLP) and stored at −80 °C.

### 4.2. Kinetic Assays

The different transamination reactions were monitored through coupled spectrophotometric assays conducted under near-physiological conditions (at pH 7.4, 37 °C) as detailed below. Kinetic data were analyzed by nonlinear least-squares fitting to the appropriate kinetic equation using Sigma Plot (Systat Software Inc. San Jose, CA, USA). In particular, substrate titration data were fitted to the Michaelis–Menten equation, or to a variation of the same that directly yields *k_cat_*/*K_M_*.
(6)$$v=\frac{e\times K_{M}\;\left[E\right]\left[S\right]}{K_{M}+\left[S\right]}$$
where v is the initial rate of the reaction, [E] is the total enzyme concentration, [S] the initial substrate concentration and e represents *k_cat_*/*K_M_*.

### 4.3. Assays of GOT1-Catalyzed Reactions

The transamination reaction between l-aspartate and α-ketoglutarate (α-KG) operated by GOT1 was monitored via a continuous coupled assay with malate dehydrogenase. Conditions were: 50 mM Hepes, pH 7.4, 37 °C, 50 mM NaCl, 0.25 mM α-KG, 0.25 mM NADH and 0.5 μM human cytosolic malate dehydrogenase (MDH1). The concentration of GOT1 was 4 nM and the concentration of l-aspartate ranged between 0.2 and 10 mM. The assay was performed in 1-cm plastic cuvettes in a thermostatted spectrophotometer and the reaction was monitored by following the absorbance at 340 nm.

The reverse of the reaction above (transamination between l-glutamate and oxaloacetate) operated by GOT1 was assayed trough a continuous coupled assay with glutamate dehydrogenase (GDH). Conditions were: 50 mM Hepes, pH 7.4, 37 °C, 50 mM NaCl, 20 mM NH_4_Cl, 0.2 mM oxaloacetate, 0.25 mM NADH and 3 U/mL GDH. The concentration of GOT1 was 4 nM and the concentration of l-glutamate ranged between 0.2 and 10 mM.

The transamination reaction between l-alanine and α-ketoglutarate operated by GOT1 was monitored through a continuous coupled assay with lactate dehydrogenase. Conditions were: 50 mM Hepes, pH 7.4, 37 °C, 50 mM NaCl, 0.25 mM α-KG, 0.25 mM NADH and 50 nM human lactate dehydrogenase (LDHA). The concentration of GOT1 was 1 μM and the concentration of l-alanine ranged between 0.8 and 40 mM.

The transamination between l-asparagine and α-KG catalyzed by GOT1 was monitored through a continuous coupled assay with ω-amidase and MDH1. MDH1 itself showed no appreciable activity towards the ketoacid generated from l-asparagine (α-ketosuccinamate). However ω-amidase (the enzyme used herein was the product of the nit3 gene from *S. cerevisiae* [36]) converted α-ketosuccinamate to oxaloacetate, which could then be reduced by MDH. The assay had to be conducted with some care to avoid artifacts. In particular, commercial l-asparagine is routinely contaminated by small amounts of l-aspartate (in the present case, the contamination was estimated to be in the order of 0.15%), which gave rise to initial apparent ‘bursts’ of activity. These initial transients were ignored in the analysis of the kinetics, and rates were estimated from the subsequent part of the reaction traces. Furthermore, ω−amidase did have some activity towards l-asparagine (yielding l-aspartate, which in turn could react with the transaminase); to limit this problem, concentration of the amidase was kept low compared to the transaminase (35 nM vs. ~1 μM) and controls in which ω-amidase was added only at the end of the kinetics served to take into account this contribution to the observed rate. The reaction mixture (1 mL final) contained 50 mM Hepes (pH 7.4) 50 mM NaCl, 0.25 mM α-KG, 0.25 mM NADH. MDH1 was 0.5 μM and yNit3 (ω-amidase) was 35 nM. The concentration of GOT1 was 1 μM and the concentration of l-asparagine ranged between 0.8 and 40 mM.

The transamination reaction between l-glutamine and oxaloacetate, catalyzed by GOT1, was assessed through a discontinuous GDH-coupled assay based on a single three-hour time point. The enzyme was incubated for three hours at 37 °C, in the presence of l-glutamine (4 to 40 mM) and 200 μM oxaloacetate in 50 mM Hepes buffer pH 7.4, 50 mM NaCl, 20 mM NH_4_Cl, 0.25 mM NADH and 25 nM ω-amidase. The final volume of the reaction mixture was 1 mL. At the end of the incubation, the samples were rapidly transferred to plastic cuvettes in the spectrophotometer and the amount of transaminated glutamine was assessed through the addition of an excess of GDH. Controls in which ω-amidase was omitted were run in parallel, to account for the presence of small amounts of contaminating l-glutamate.

### 4.4. Assays of GPT-Catalyzed Reactions

GPT activity in the transamination reaction between l-alanine and α-KG was monitored using a continuous assay based on lactate dehydrogenase. Conditions were: 50 mM Hepes, pH 7.4, 37 °C, 50 mM NaCl, 0.25 mM α-KG, 0.25 mM NADH and 0.5 μM LDHA. The concentration of GPT was 4 nM and the concentration of l-alanine ranged between 0.2 and 10 mM.

The reverse of the reaction above (transamination between l-glutamate and pyruvate) operated by GPT was assayed trough a continuous coupled assay with glutamate dehydrogenase (GDH). Conditions were: 50 mM Hepes, pH 7.4, 37 °C, 50 mM NaCl, 20 mM NH_4_Cl, 0.25 mM NADH and 3 U/mL GDH. The concentration of GPT was 4 nM. When the substrate to be varied was l-glutamate, the concentration of pyruvate was kept fixed at 0.25 mM and the amino acid ranged between 0.2 and 10 mM. Conversely, when the substrate to be varied was pyruvate, l-glutamate was kept fixed at 5 mM and pyruvate ranged between 0.1 and 5 mM.

The transamination between l-glutamate and α-ketobutyrate was again assayed trough a continuous coupled assay with GDH. Conditions were: 50 mM Hepes, pH 7.4, 37 °C, 50 mM NaCl, 20 mM NH_4_Cl, 0.25 mM pyruvate, 0.25 mM NADH and 3 U/mL GDH. The concentration of GPT was 8 nM and the concentration of α-ketobutyrate ranged between 0.2 and 10 mM.

The transamination between glycine and α-KG, catalyzed by GPT, was measured using a continuous assay based on lactate dehydrogenase. Conditions were: 50 mM Hepes, pH 7.4, 37 °C, 50 mM NaCl, 0.25 mM α-KG, 0.25 mM NADH and 0.5 μM LDHA. The concentration of GPT was 8 nM and the concentration of glycine ranged between 5 and 60 mM.

The transamination between l-glutamine and pyruvate catalyzed by GPT was monitored through a discontinuous coupled assay with ω-amidase and MDH1. The assay was performed at 37 °C in 1-cm plastic cuvettes. The reaction mixture (1 mL final) contained 50 mM Hepes (pH 7.4) and 50 mM NaCl. MDH1 was 0.4 μM and yNit3 (ω-amidase) was 20 nM. Glutamine ranged between 0.5 and 30 mM, while the concentration of α-KG was 0.5 mM.

### 4.5. Literature Searches

To put the *D* values established for GOT1 and GPT into context, we performed a literature survey of experimentally determined selectivities, focusing on cases where enzymes had to discriminate between substrates showing limited structural/chemical differences.

For each particular difference considered (e.g., the substitution of a carboxylate with an amido group), the search proceeded stepwise. First a list of pairs of compounds was drawn, in which the two members of a pair differed only by the feature of interest (e.g., l-aspartate/l-asparagine; l-glutamate/l-glutamine; nicotinate/nicotinamide and so forth). Then, the scientific literature was thoroughly searched (using the BRENDA repository as a starting point [37], but integrating it with searches on Google Scholar and other literature databases) to identify enzymes known to react with both substrates in a pair. The following criteria were adopted:

(a)The enzyme had to have been tested against the two substrates in the same study and the catalytic parameters for the two substrates had to have been obtained under otherwise identical conditions. (Note that in some cases, although the enzyme name designates a particular compound as the substrate, the preferred substrate was in effect another.)(b)The chemical group that was different between the two substrates should not be involved directly in the reaction nor change substantially the chemical properties of an adjacent group undergoing reaction (e.g., change an aldehyde into a ketone).(c)If, in the same study, selectivity was assessed under more than one condition (e.g., at different pH values, using different, physiologically available co-substrates, etc.), only the condition yielding the highest discrimination index was taken into account. In case the kinetics of the enzyme under examination showed cooperativity, an approximate discrimination index was calculated based on the ratio of *k_cat_/S_0.5_* values for the two substrates, even though this procedure is not entirely correct [8].

Depending on the way reaction rates were expressed, sometimes the original literature did not report proper values of *k_cat_*/*K_M_*, but only values of *V_max_/K_M_* (or separate values for *V_max_* and *K_M_*, from which the ratio could be computed). In these cases, irrespective of the units in which *V_ma_*_x_ was expressed, the discrimination index *D* was calculated from the ratio of *V_max_/K_M_* values for the two substrates, relying on the fact that *V_max_* is directly proportional to *k_cat_*. In one case, the index was estimated directly from plots presented in the original article [38]. 

### 4.6. Molecular Docking Simulations

The docking simulations were performed with the Autodock 4.2 software package, after preparing the input files with the aid of AutoDockTools (ADT) interface [39]. The Lamarckian Genetic Algorithm [40] was used for all docking calculations. We performed 1000 docking runs, each with 5 × 10^6^ energy evaluations, 200 individuals in the initial population, and 27,000 generations, assuring a convergence of the docking. A conformational cluster analysis was performed on the docked conformations [40] with a rmsd cluster threshold of 2 Å and structural analysis of the binding modes was made using ADT, VMD [41] and Swiss-PdbViewer [42] softwares.

In the structure of human GOT1, retrieved from the Protein Data Bank [43] (PDB ID: 3II0), the asymmetric unit contains two dimers (chains A-D and B-C). Each monomer is complexed with the cofactor pyridoxal 5′-phosphate (PLP) and tartaric acid. Some sidechains are unresolved, but fortunately they were far from the binding site and were reconstructed by means of the Swiss-PdbViewer software [42]. The dimer formed by chains A and D was chosen for the docking simulations due to the lowest number of unresolved/reconstructed sidechains, and it was used after deleting all ligands and water molecules. The binding sites of all monomers were perfectly superimposable and the one of chain D was selected for docking. The grid box for affinity maps was indeed centered in the binding site, with a dimension of 64 × 72 × 62 points and a spacing of 0.375 Å. 

Histidine residues were kept in the neutral form. The Lys259 sidechain was also kept in the neutral form to simulate the real protonation state of this intermediate. 

The structures of different substrates (l-aspartate, l-glutamate, l-asparagine and l-glutamine) complexed to the PLP cofactor bound via a Schiff base were built by means of PRODRG server [44]. After a check of the obtained conformation and of the partial charges assigned to the atoms, the all-hydrogens structure was imported in ADT without further modifications except the merging of non-polar hydrogens. All the rotatable bonds were kept free to rotate.

### 4.7. GPT Modelling

Models of human cytosolic alanine aminotransferase were retrieved both from the Swiss-Model repository [45,46] and from the Alpha-Fold Database [47,48].

The two models built by Swiss-Model both contain the PLP cofactor. The first one is based on the structure of human mitochondrial aminotransferase (69.26% sequence identity), whose asymmetric unit is monomeric (PDB ID:3IHJ). It has a very high average model confidence [49] (QMEANDisCo = 0.86), with the exception of two loop regions (residues 68–77 and 116–124) for which the value decreases until about 0.4. The first of these regions is located near the binding site. The second model is based on the structure of alanine aminotransferase from *Hordeum vulgare* (46.95% of sequence identity, PDB ID: 3TCM) and it is in a dimeric form. It has a high model confidence too (QMEANDisCo = 0.78), with the exception of three small loop regions far from the binding site (residues 94-100, 175–177 and 299–302), for which the value decreases to about 0.4. 

The model predicted by AlphaFold is in a monomeric form and it has again a very high confidence [50] (pLDDT > 90), with the exception of the N-terminal region (residues 1–14), for which the value decreases till about 0.4.

The structural comparison of the three models shows a very good superimposition of the second model of Swiss-Model with the AlphaFold model, assessed by the “consistency with ensemble” score, that identifies local deviations of a protein structure from the ‘consensus’ established by all other structures selected for comparison. The first model of Swiss-Model is the worst, in particular concerning the N-terminus and the loop near the binding site.

Because the dimeric form is necessary to have the complete set of key residues of the binding site, the second model built by Swiss-Model was selected for further analysis, described in the Results.

## Data Availability

The data presented in this study are available on request from the author.

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
