# Peer review of "Using Steady-State Kinetics to Quantitate Substrate Selectivity and Specificity: A Case Study with Two Human Transaminases"

_molecules, 2022, doi:10.3390/molecules27041398_

Round 1
Reviewer 1 Report
This is a reworked version of a manuscript I have earlier reviewed. It does not tread much new ground, but it is an improvement over the earlier version and may be interesting for researchers on this area
Issues:
A)There is one instance of confusing vocabulary in the introduction: in lines 57-70, it appears that specificity is kcat/Km , and later selectivity seems to be defined as D (the ration of kca/Km for both substrates of interest). Later, though, the authors state: " However, while selectivity may be quantitated quite satisfactorily through the D index, there is no accepted synthetic measure of specificity". It this appears that there is some contradiction between this sentence and the earlier text in lines 57-70
B) line 169: authors state "it can be formally shown that
kcat/KM is directly related to the free energy of interaction between the substrate and the enzyme in the transition state [16];" This is not correct: kcat/Km is related to the DIFFERENCE in energies of interaction between substrate and active site between transition state and enzyme-substrate complex PROVIDED that the catalytic reaction is much faster than an unproductive unbinding event. I called the authors' attention to the flaws in the their reading of ref. 16 a previous review, and I am disappointed to see that this is still not correct.
C) Regarding the values in Tables 1 and 2 authors state "kcat/KM represents the slope of the initial part of the Michaelis-Menten hyperbola (reaction rate vs. substrate concentration). Therefore, even for those enzyme-substrate couples in which substrate saturation cannot be achieved due to poor binding (hence preventing a reliable measurement of individual values for kcat and KM) one can obtain a reliable estimate of the kcat/KM ratio." I therefore assume that the kcat/Km ratios have been computed from such graphical representations. However, current numerical methods of equation fitting would surely allow them to obtain individual estimates of both kcat and KM even for cases of weak binding. The paper would be improved if those values were included in tables 1/2
D) line 183: authors state ". As it can be seen, L-asparagine, alanine and L-glutamine were all discriminated by the enzyme around 10^6 fold. " This sentence is ambiguous. " As it can be seen,the enzyme discriminates against L-asparagine, alanine and L-glutamine by around 10^6 fold.
E) lines 290-300 : The explanation of the deficient catalysis of asparagine/glutamine does make sense, but a better explanation of the origin of better docking energies for the rotated conformation in asparagine and glutamine is lacking. Additionally, a better estimate of the binding energies in both modes would be desireable since if both modes were (as the text says) equally favored one would expect the reaction rate to only drop to one half (since 1/2 of the molecules would still bind in the correct orientation, where proton trasnfer from Lys is possible). In Fig. 5, the histograms in the second and third rows seem to have been mixed up.
F) The section in the GPT modeldoes not contribute anything to the paper, since it does not explicitly clarify the origin of the discrimination agains glycine, aspartate or glutamine.
Author Response
Referee 1
This is a reworked version of a manuscript I have earlier reviewed. It does not tread much new ground, but it is an improvement over the earlier version and may be interesting for researchers on this area
Issues:
A) There is one instance of confusing vocabulary in the introduction: in lines 57-70, it appears that specificity is kcat/Km , and later selectivity seems to be defined as D (the ration of kca/Km for both substrates of interest). Later, though, the authors state: "However, while selectivity may be quantitated quite satisfactorily through the D index, there is no accepted synthetic measure of specificity". It this appears that there is some contradiction between this sentence and the earlier text in lines 57-70
Reply: The source of confusion arises, we believe, from the denomination of kcat/KM as the ‘specificity constant’. Such a denomination is standard in the literature, but somewhat misleading, as evidently specificity (and selectivity) cannot be inferred from a parameter obtained with a single substrate, but must entail comparisons between different substrates. This need for comparisons is stated clearly in the introduction, and reprised even more explicitly in the discussion. To address the ambiguity noted by the referee, we now avoid referring to kcat/KM as the ‘specificity constant’, both in the introduction and throughout the text.
B) line 169: authors state "it can be formally shown that kcat/KM is directly related to the free energy of interaction between the substrate and the enzyme in the transition state [16];" This is not correct: kcat/Km is related to the DIFFERENCE in energies of interaction between substrate and active site between transition state and enzyme-substrate complex PROVIDED that the catalytic reaction is much faster than an unproductive unbinding event. I called the authors' attention to the flaws in the their reading of ref. 16 a previous review, and I am disappointed to see that this is still not correct.
Reply: To address the point signaled by the referee we have now included a new appendix (Appendix A in the revised manuscript) to provide support for our statements – instead of referring to previous papers. We hope that the treatment, schemes and formulas presented in this appendix may adequately explain the point we were trying to make in the main text.
C) Regarding the values in Tables 1 and 2 authors state "kcat/KM represents the slope of the initial part of the Michaelis-Menten hyperbola (reaction rate vs. substrate concentration). Therefore, even for those enzyme-substrate couples in which substrate saturation cannot be achieved due to poor binding (hence preventing a reliable measurement of individual values for kcat and KM) one can obtain a reliable estimate of the kcat/KM ratio." I therefore assume that the kcat/Km ratios have been computed from such graphical representations. However, current numerical methods of equation fitting would surely allow them to obtain individual estimates of both kcat and KM even for cases of weak binding. The paper would be improved if those values were included in tables 1/2
Reply: As expounded in the Methods, our data were analyzed by nonlinear least-squares fitting to the Michaelis-Menten equation, or to a variation of the same that directly yields kcat/KM. Such fittings allow one “to obtain individual estimates of both kcat and KM” (as indicated by the referee). However, for weakly binding substrates these estimates of kcat and KM may contain huge errors or even make no physical sense at all. For example, the computer-estimated KM for GPT reacting with L-Asp (at the highest concentration of 10 mM) was in the order of 5000 M! This is because when fitting a nearly-linear dependence of rate vs. [substrate] to a hyperbola, small errors in the data would translate into substantial changes in the estimates of kcat and KM. Accordingly, for such weakly binding substrates, we only provide a lower limit for KM (corresponding to the highest concentration of substrate used).
On the other hand, we maintain that the kcat/KM ‘ratio’ can be estimated very reliably from our fittings, as the parameter mirrors the slope of the initial part of the Michaelis-Menten hyperbola. Indeed, for weakly binding substrates, essentially identical kcat/KM values could be obtained by fitting the experimental data to a straight line passing through the axes’ origin.
The reliability of kcat/KM values obtained in the absence of substrate saturation was (and still is) is explained in detail on page 4 of the text. In the revised manuscript, we have added a footnote to Tables 1 and 2, to reiterate the point.
D) line 183: authors state ". As it can be seen, L-asparagine, alanine and L-glutamine were all discriminated by the enzyme around 10^6 fold. " This sentence is ambiguous. " As it can be seen,the enzyme discriminates against L-asparagine, alanine and L-glutamine by around 10^6 fold.
Reply: for the sake of clarity, we have rephrased the sentence signaled by the referee.
E) lines 290-300 : The explanation of the deficient catalysis of asparagine/glutamine does make sense, but a better explanation of the origin of better docking energies for the rotated conformation in asparagine and glutamine is lacking. Additionally, a better estimate of the binding energies in both modes would be desireable since if both modes were (as the text says) equally favored one would expect the reaction rate to only drop to one half (since 1/2 of the molecules would still bind in the correct orientation, where proton trasnfer from Lys is possible). In Fig. 5, the histograms in the second and third rows seem to have been mixed up.
Reply: We thank the referee for pointing up the mixup in the figure, which has now been corrected (Figure 6 in the revised version). Regarding the (calculated) binding energy of the rotated conformation, this energy is higher than for the canonical conformation only in the case of glutamine, and the difference is negligible anyway (much like for asparagine). Finally, the presence of an alternate conformation is only part of the explanation for the lower reactivity of Gln and Asn: two more important factors seem to be the weakened binding energy (for the canonical, and hence presumably productive, binding mode) with respect to Glu and Asp, and the appreciable changes in the orientations of the alpha carbon for the amido-containing substrates (a feature which may be affecting kcat). These factors are summarized on page 10.
F) The section in the GPT model does not contribute anything to the paper, since it does not explicitly clarify the origin of the discrimination agains glycine, aspartate or glutamine.
Reply: In our opinion, building a structural model for GPT has made it possible to envisage at atomic level some interesting structural aspects of the active site, never described before, and to relate them with the specificity of GPT for different substrates. First and foremost, the model shows the presence in the binding site of two residues with different physico-chemical properties (tyrosine and leucine), able to favorably interact with different substrate sidechains (Figure 8, appendix C).
The validity of the modelled GPT active site, is confirmed by the conserved key residues (e.g. the arginines that keep the phosphate group and the alpha-carboxylate), whose position and orientation are preserved along homologous protein structures, as well as by preserved key interactions (e.g. the tyrosine that replaces tryptophan in stacking with PLP and in the H-bond with the substrate sidechain), which help to maintain the enzyme functionality.
In the revised version, the points made above are better highlighted in the Result and Conclusion. Figure 8 in Appendix C has also been modified to better illustrate our points.

Reviewer 2 Report
The manuscript from Perecchi and Polverini deals with the evaluation of substrates specificity and selectivity of human transaminases. The article discusses an interesting topic, is well written and structured.
The new version of the manuscript takes into account most of my previous comments, and in addition the authors have complemented and deepened their discussions with enzyme docking and structural homology analysis.
Overall, the quality of the work is outstanding, and my recommendation is to accept the manuscript in its current form.
Author Response
Referee 2:
The manuscript from Perecchi and Polverini deals with the evaluation of substrates specificity and selectivity of human transaminases. The article discusses an interesting topic, is well written and structured.
The new version of the manuscript takes into account most of my previous comments, and in addition the authors have complemented and deepened their discussions with enzyme docking and structural homology analysis.
Overall, the quality of the work is outstanding, and my recommendation is to accept the manuscript in its current form.
Reply: We thank the referee for his/her appreciation.
Round 2
Reviewer 1 Report
I am satisfied by the changes introduced.